# Evaluation of Chelating Agents Used in Phytoextraction by Switchgrass of Lead Contaminated Soil

**DOI:** 10.3390/plants11081012

**Published:** 2022-04-08

**Authors:** Genna Hart, Marina Koether, Thomas McElroy, Sigurdur Greipsson

**Affiliations:** 1Department of Ecology, Evolution and Organismal Biology, Kennesaw State University, 370 Paulding Ave., Kennesaw, GA 30144, USA; ghart14@kennesaw.edu (G.H.); tmcelroy2@kennesaw.edu (T.M.); 2Department of Chemistry and Biochemistry, Kennesaw State University, 370 Paulding Ave., Kennesaw, GA 30144, USA; mkoether@kennesaw.edu

**Keywords:** bioenergy-crop, EDTA, lead (Pb), NTA, phytoextraction, switchgrass (*Panicum virgatum* L.)

## Abstract

Soil lead (Pb) contamination is a recognized environmental and global health problem. Phytoextraction of Pb using switchgrass (*Panicum virgatum* L.), a second-generation biofuel crop, is typically enhanced by soil chelation. The effectiveness of four different chelating agents, phytic acid (inositol hexaphosphate), citric acid, NTA (nitrilotriacetic acid), and EDTA (ethylenediaminetetraacetic acid) was examined in pot culture. Plants treated with EDTA (1 mM) showed significantly higher shoot Pb concentrations compared to control plants and plants treated with other chelates. Lead-solubility following phytoextraction was examined by soil washing using 0.01 and 0.05 M acetic acid as an extractant solution revealed no significant differences in Pb concentrations in soil among different chelate treatments and control. Furthermore, the effects of different concentrations (1, 2, 5 and 10 mM) of NTA on Pb phytoextraction of switchgrass were examined. Plants receiving 5 mM and 10 mM NTA had significantly higher foliage concentrations of Pb compared to plants treated with lower levels (1 and 2 mM) of NTA. Moreover, the effect of NTA application alone was significantly improved by a combined application of Triton X-100, an alkyl polyglucoside (APG); the Pb concentration in the foliage of switchgrass was more than doubled when treated with NTA combined with APG. The use of NTA combined with APG has great potential in improving phytoextraction efficiencies of switchgrass on Pb-contaminated soils.

## 1. Introduction

Soil lead (Pb) contamination is mainly derived from industrial sources [1], agricultural pesticides [2], chemical fertilizers [3] and from emission of tetraethyllead ((CH_3_CH_2_)_4_Pb) gasoline [4,5,6,7]. Environmental Pb contamination is a recognized global health problem and it is well known that long-term exposure to low-level Pb can result in neurological dysfunctions including intellectual impairment in children [8,9,10], Alzheimer’s [11,12] and Parkinson’s diseases [13].

Lead (Pb) is persistent in the topsoil but can become re-suspended, resulting in Pb exposure [14,15]. Lead (Pb) soil-contamination level and blood lead level (BLL) in children are strongly correlated [14,16]. Currently, there is no BLL level known that is not considered harmful to human health [17].

The removal of Pb-contaminants from soil can be employed through phytoextraction, an emerging cost-effective heavy metal remediation technique [18]. Phytoextraction of Pb-contaminated soil involves soil acidification or chelation, which in turn facilitates the absorption and sequestration of Pb-contaminant in plant tissues [18,19]. Lead ions are positively charged, causing them to bind to particles in the topsoil and resist leaching into deeper soil horizons [20]. Thus, low bioavailability of metals in soil is often a limiting factor in phytoextraction [19,21].

Ethylenediaminetetraacetic acid (EDTA) is regarded as a highly effective soil Pb-chelate [22,23], though its long soil persistence time and its resistance to breakdown by soil microbes results in potential concerns about Pb mobilization through the soil profile [24,25,26]. Lead in the soil is distributed in several fractions with different relative mobilities [27]. Chelation by EDTA has been shown to reduce Pb in the oxide bound fraction of soil which usually includes anthropogenic Pb sources [27,28]. Another problem associated with the use of EDTA on Pb-contaminated soil is the increased mobility of macronutrients (Ca, Mg, P) and total water-soluble micronutrients (Cu, Fe, Mn) [25,29,30].

Alternative chelates to EDTA have included citric acid (C_6_H_8_O_7_) a natural low molecular weight organic acid that has been used effectively at enhancing Pb phytoextraction [31,32]. Citric acid lowers soil pH and the ideal soil pH was found between 4.0 to 4.5 for maximum Pb mobility [32]. Citric acid degrades within a week (half-life 1.5 to 5.7 days) in soil [33] which is advantageous over other chemical chelates such as EDTA that are much more persistent in the soil [31]. Citric acid (60 mM) was found to be more effective than EDTA (5 mM) and NTA (5 mM) in increasing Pb phytoextraction of multi metal contaminated sediment by castor bean (*Ricinus communis* L.) and chicory (*Cichorium intybus* L.) [34]. Citric acid (25 mmol kg^−1^ soil) can be effectively used to increase Pb uptake of switchgrass (*Panicum virgtum* L.) [32]. However, acidifying soil to pH 4.0 using citric acid resulted in aluminum (Al) toxicity of switchgrass [32]. This was a considerable drawback of using citric acid in phytoextraction of Pb.

The Pb-chelate NTA has been suggested as an alternative to EDTA [35]. The NTA is a derivative of EDTA and has been shown to have similar chelating effects [35]. The main advantage of using NTA in phytoextraction is how fast it is biodegraded in soil in comparison to EDTA that is non-biodegradable [36]. It has been previously established that in soils, NTA was readily decomposed principally by microorganisms under aerobic conditions with half-lives ranging from 3 to 7 days [37,38]. Moreover, the use of NTA presents lower toxicity to microorganisms and plants [19]. Phytoextraction efficiency of Pb-contaminated soil by corn (*Zea mays* L.) was found to be greater when using EDTA compared to NTA but leaching of Pb through the soil column was doubled using EDTA compared to NTA [39]. The efficiency of NTA application in phytoextraction of Pb-contaminated soil by *Scirpus triqueter* was improved by combined application of alkyl polyglucoside (APG) [35]. The NTA is seen to be a possible alternative to EDTA since NTA is more readily biodegradable in soil compared to EDTA and is more economically available due to its commercial mass use as a phosphate free detergent [40]. Chelation of Pb in soils by combining the natural processes of switchgrass with the manipulation of soil chemistry using organic or aminopolycarboxylic acids has been shown to increase the bioavailability of Pb [32,41,42]. Other promising chemicals such as chitosan and humic substances have recently been shown to increase the bioavailability of Pb [43,44].

Switchgrass is regarded as a second-generation biofuel crop and its biomass is used in advanced biofuel production [45,46]. Phytoremediation of contaminated soils using second-generation bioenergy crop such as switchgrass has great potentials [47,48,49,50]. The cultivation of biofuel crops on marginal lands may improve energy security and aid in mitigating climate change [51]. In addition, the cultivation of bioenergy crops on marginal lands may reduce the need for using primary agricultural lands for biofuel production. 

The main objective of this study was to examine the efficacy of several chelating agents as a potential alternative to EDTA chelation in phytoextraction by switchgrass of Pb-contaminated soil.

## 2. Results

### 2.1. Effect of Different Chelates on Phytoextraction of Switchgrass and Soil Washing

Chemical analysis of Pb concentrations in plant foliage revealed significant differences in Pb concentrations of harvestable biomass among different chelating treatments. Plants treated with EDTA significantly showed the highest shoot Pb concentrations compared to plants in other treatments (Figure 1). Plants treated with other chelates (citric acid, phytic acid and NTA) did not have a significantly different concentration of Pb in the foliage compared to control plants (Figure 1). The potential of different chelating agents to increase available Pb by soil wash was examined. Analysis of Pb concentration in both 0.01 M and 0.05 M acetic acid soil wash revealed no significant differences between different treatments (Figure 2a,b).

### 2.2. Effect of Different Levels of NTA on Pb Phytoextraction of Switchgrass

Challenging plants with different levels of NTA clearly showed different responses in Pb accumulation and growth of plants. Plants receiving 10 mM NTA showed clear toxic Pb effects, such as chlorosis. Although the dry mass of plants did not differ significantly between treatments, the average Pb concentration of the foliage differed significantly between treatments (Figure 3a,b). Plants treated with 5 mM and 10 mM NTA had a significantly higher average concentration of Pb in the foliage than plants that were treated with 1 and 2 mM NTA (Figure 3a). Increasing the concentration of NTA from 1 mM to 5 mM greatly impacted plant’s Pb uptake; plants treated with 5 mM NTA had an almost 800% higher Pb concentration in the foliage compared to plants receiving 1 mM NTA (Figure 3a). Elevating the NTA application to 10 mM NTA did not result in a significant increase in Pb concentration of the foliage or biomass production compared to plants that received 5 mM NTA.

### 2.3. Effect of NTA and Combined Application of APG or Nano-Silica on Pb Phytoextraction

The efficiency of the NTA chelate in the enhancing uptake of Pb by switchgrass was significantly improved by combining the NTA (5 mM) with the Triton X-100. Plants that received soil application of the combined NTA with the APG significantly had the highest foliage concentration of Pb (Figure 4a). The Pb in the foliage of plants treated with the combined NTA and APG was more than double that of plants that were treated with NTA alone (Figure 4a). The iron (Fe) concentration in the foliage of plants did not differ between treatments, indicating no impact on plant’s Fe homeostasis (Figure 4b). The Fe concentration in leaves increased stepwise between treatments (Figure 4b). A linear regression of treatment categories (see Figure 4b) vs. foliage Fe concentrations showed a strong relationship (R^2^ = 0.99) between the average Fe concentration in leaves and treatments. Plants that received soil application of NTA alone or in a combined application of NTA and APG had a significantly higher foliage concentration of aluminum (Al) (Figure 4c). Plants received a combined application of NTA and nano-silica, indicating that the nano-silica application could potentially inhibit excess uptake of Al (Figure 4c).

## 3. Discussion

This study examined the efficiency of several chelating agents used in phytoextraction by switchgrass (*Panicum virgatum* L.) and their impact on Pb solubility in soil. The in situ use of the synthetic chelate EDTA has met criticism due to possible negative effects associated with long persistence in soil, leading to the potential mobilization of Pb through the soil profile and negative effects on plant growth [24,26]. Previous studies have showed that EDTA treatments increased Pb accumulation in plants [32,42]. However, EDTA is persistent in soil and may mobilize Pb and other metals through the soil profile and into groundwater, thus increasing the risk of human exposure [24,26,36,52]. In addition, EDTA has been observed to negatively impact plant health and reduce biomass [53,54]. Therefore, natural acids and other chelates with shorter half-lives in soil are being scrutinized as alternatives to EDTA. This study showed that treating soil with different chelating agents improved Pb-uptake by switchgrass from Pb-contaminated soil. Although using EDTA (0.5 mM) resulted in a significantly higher Pb concentration in the foliage of switchgrass, the phytoextraction efficiency of NTA was further improved at higher doses (5 mM). At the same time, the dry mass of plants in different NTA treatments remained slightly less but non-significantly different. However, it must be considered that phytoextraction efficiency of Pb is a function of plant’s dry mass and concentration of Pb in the foliage [18]. Therefore, this study indicates that phytoextraction of Pb contaminated soil can be improved by using 5 mM NTA. Other studies have recommended the use of NTA in phytoextraction of Pb-contaminated soil given the potential environmental risks of EDTA [55]. Furthermore, phytoextraction efficiency of EDTA (0.5 mM) and NTA (0.5 mM) were previously found to be similar for three plant species grown in high levels of Pb contaminated soil [56].

The split dose (i.e., repetitive application of a chelate to the same plant) treatment has been suggested as an alternative method for the use of EDTA [27,57], whereas no significant difference in Pb phytoextraction were observed by *Dianthus chinensis* between the treatments of adding 5 and 10 mmol kg^−1^ of EDTA [58]. Moreover, split doses of EDTA at 2.5 mmol kg^−1^ was found to be more effective for Pb accumulation in shoots of buckwheat (*Fagopyrum esculentum*) and sunflowers (*Helianthus annuus*) than a single dose of 5.0 mmol kg^−1^ [59]. The selection of an appropriate chelating agent might depend on the soil type, as citric acid has, under certain conditions, been selected as a preferred chelating agent over EDTA or NTA [60]. Citric acid has been proposed as a natural chelation acid because it is readily broken down by microbes, giving it short soil persistence and reducing the risk of metal leaching into groundwater [31]. Citric acid does require multiple applications to effectively reach and maintain the ideal soil acidity (pH 4.0–4.5) to keep Pb available for plant uptake over an extended period [31,32]. However, acidifying soils to such a low level can have other adverse effects. Citric acid application reduced soil pH to 4.0, which in turn resulted in excessive aluminum (Al) and iron (Fe) uptake of switchgrass, resulting in toxic effects [32]. This study showed that although the NTA + APG application resulted in the highest Pb accumulation in the foliage of switchgrass the Al and Fe levels were low and did not cause toxicity in the plants. This study showed that when nano-silica was added to the NTA solution, it resulted in significantly less concentration of Al concentration in plant’s foliage compared to when plants received NTA alone. These findings warrant further study on the combined application of nano-silica and citric acid in phytoextraction. A previous study suggested that combined citric acid and soil fungicide application could achieve similar results as EDTA application [32]. The application of citric acid has previously not shown any clear and significant pattern with regards to plant’s biomass reduction [41]. However, a synergistic effect of the fungal suppressants, propiconazole especially, and citric acid, may have reduced the biomass of switchgrass and resulted in significantly negative growth effects [41].

This study demonstrated that a low level of EDTA chelation does not result in excessive Pb concentrations in the soil wash compared to other chelating agents such as NTA, citric acid and phytic acid. The lack of significant differences in Pb concentrations of the soil wash of different treatments suggests that treating soil with chelates at low concentrations to enhance Pb uptake of plants does not necessarily result in excessive Pb-solubilization and potential Pb-migration through the soil profile. Other studies have found that leaching of Pb through the soil column was doubled by the use of EDTA using higher concentrations (10, and 20 mmol kg^−1^) compared to equimolar solutions of NTA [39].

Phytoextraction efficiency can also be optimized by augmenting plant’s biomass, which is also important for the bioenergy industry. A recent study did not find a significant reduction in biomass of switchgrass treated with a single dose of EDTA compared to the control plants [41]. Plant’s biomass can be augmented by adding soil nitrogen (1000 mg kg^−1^ soil), which also resulted in an improved uptake of Pb by switchgrass (69). Similarly, foliar application of iron (Fe) (20 mg kg^−1^ solution) resulted in significantly higher biomass of the foliage compared to control plants [61]. The use of plant growth regulators has also been advocated to increase phytoextraction efficiency. Application of the plant growth regulator benzylaminopurine (BAP) on switchgrass was found to significantly increase biomass and shoot Pb concentrations compared to control plants [32]. Other plant growth regulators, such as gibberellic acid (GA_3_), in combination with sodium nitroprusside (SNP) and diethyl aminoethyl hexanoate (DA-6) and salicylic acid (SA), have shown improved biomass production of switchgrass [41,62]. Previous studies have shown that various chemical manipulations enhanced Pb phytoextraction. Previous studies have demonstrated that suppression of arbuscular mycorrhizal fungi (AMF) activity through the use of the soil fungicide benomyl (C_14_H_18_N_4_0_3_) resulted in an increased accumulation of Pb into the harvestable foliage of corn (*Zea mays*) and switchgrass [63,64]. Furthermore, when benomyl application was combined with EDTA, the foliage Pb concentration increased more than 900% when compared to control plants [42]. The phytoextraction of Pb by switchgrass enhanced with chelate applications has many implications in future research for both the phytoremediation and bioenergy industries.

## 4. Materials and Methods

### 4.1. Plant Species

Switchgrass (*Panicum virgatum* L.) is a C4 perennial grass adapted to a broad range of climates, topography and soil conditions throughout North America [65,66]. Switchgrass has high biomass production, and the variety “Alamo” can generate 17,800 kg of harvestable tissue per hectare [67]. Switchgrass may be harvested more than once in a growing period, and as a perennial grass it will continue to grow for up to 10 years [68]. The cost of this biomass production is estimated to be much lower than other high biomass yield crops [69]. Another attribute contributing to selection of switchgrass is its tolerance for Pb in soils [70]. The switchgrass variety “Alamo”, that is particularly well acclimated for use in Georgia, was used in this study [71].

### 4.2. Soil

Soil was collected from a highly contaminated former Superfund site in Cedartown, GA. In a final report from 2006 on the Cedartown Superfund site made by ASTR (Agency for Toxic Substances and Disease Registry) soil samples with extremely high Pb concentrations (up to 260,000 mg kg^−1^) were still found. The soils of Georgia are ultisols that are generally clay-rich, acidic and have a low base cation saturation [72]. The ultisols are rich in quartz, feldspar, mica, Fe-oxyhydroxides, kaolinite, and illite [73]. Soil samples (*n* = 25) were collected at randomly selected intervals along a 25 m long transect using a standardized soil corer (20 cm deep and 10 cm in diameter). This resulted in a total of 75 soil samples. The soil samples were transported in a cooler and were thoroughly mixed. Pots (5 L) were filled with the Pb contaminated soil. The soil was left unsterilized to maintain the indigenous soil microbiota; however, debris larger than 0.5 cm were removed by hand. Chemical analysis (by the Soil, Plant, and Water Laboratory of the College of Agricultural & Environmental Sciences, University of Georgia, 2400 College Station Road, Athens, GA 30602) demonstrated that the soil contained high concentrations of lead (Pb) (5802.5 mg kg^−^^1^), iron (Fe) (12,316 mg kg^−1^) and aluminum (Al) (5362 mg kg^−1^) (57). In addition, eleven species of arbuscular mycorrhizal fungi (AMF) were identified in the soil from Cedartown using DNA metabarcoding, a high-throughput, taxonomic identification of micro-community assemblages within a soil sample [61].

### 4.3. Effect of Different Chelates on Phytoextraction of Switchgrass

The effectiveness of four different chelates, EDTA, NTA, phytic acid (C_6_H_18_O_24_P_6_) and citric acid were tested for lead (Pb) phytoextraction of switchgrass. In addition, plants were also grown without any chelate addition (control). Twenty-five pots (5 L) were filled with soil collected from Cedartown (see above). The soil was left unsterilized to maintain the indigenous soil microbiota; however, debris larger than 0.5 cm were removed by hand. Fifteen seeds of switchgrass were seeded in each pot at a depth of 0.25 cm. Plants were subjected to four different experimental treatments: (1) control; (2) citric acid (10 mM) (CA); (3) phytic acid (10 mM) (PA); (4) NTA; and (5) EDTA. Five replicated pots were used per treatment. An equimolar chelate concentration for NTA and EDTA was 1 mmol kg^−1^ of soil. Plants were grown under controlled environmental conditions in the Science Greenhouse at Kennesaw State University (KSU), Kennesaw, GA, USA, at an average temperature of 25 °C. Humidity was monitored and maintained at a near-constant level. The pots were placed on wire-topped greenhouse benches with individual plastic saucers placed under each pot to prevent soil loss and cross contamination. Natural light varied over time but not across treatments, with the sun availability as per the greenhouse conditions and supplemented with 14 h of artificial cool-white-fluorescent overhead light (10,000 Lux) each day. Plants were harvested at 70 days after planting (dap). Plant foliage was dried in an oven at 65 °C for 48 h prior to acid digestion.

### 4.4. Soil Washing Experiment

Soil from all treatments (see above) was stored in the lab for further analysis. After storage for 10 months, the mobile fraction of Pb in the soil of the different chelate treatments (described above) was analyzed. Changes in pH by addition of acids affects the Pb in the most easily mobilized fractions, the exchangeable and carbonate fractions [27,74]. Acetic acid (0.05 M and 0.01 M) extraction was used to recover the Pb fraction of the pots of each chelate treatment and was air dried and sieved to less than 2 mm through a stainless-steel mesh. Acetic acid (50 mL) was added to the soil samples from each treatment. The solution was shaken at a speed of 150 rpm for 1 h and then filtered through 0.45 micron filters before being chemically analyzed.

### 4.5. Effect of Different Levels of NTA on Pb Phytoextraction of Switchgrass

Seeds of switchgrass were germinated on a petri dish with filter papers layered under and over the seeds and moistened in sterilized DI water. The petri dishes were tightly closed and left for one week on a lab bench. Healthy seedlings of switchgrass were transplanted into 300 mL plastic grow tubes (*n* = 15) filled with the Pb-contaminated soil described above. Each grow-tube received one pre-germinated seedling of switchgrass. Plants were given 50 mL of complete nutrient solution twice a week. Grow tubes were placed on wire-topped greenhouse benches. Grow-tubes were arranged in a randomized block design. Each week, grow tubes were re-randomized on the benches. The plants were grown in the KSU research greenhouse under controlled environmental conditions. Plants were supplemented with cool-white, fluorescent light for 16 h per day. Temperature (average 25 °C) and humidity was monitored and maintained at a near-constant level. The tubes were given complete nutrient (332 mg L^−1^ N, 111 mg L^−1^ P, and 222 mg L^−1^ K) solution (50 mL) three times per week. At 64 days after planting (dap), the soil fungicide propiconazole (2 mg L^−^^1^) (trade name Infuse^®^) was applied (50 mL) to suppress symbiotic arbuscular mycorrhizal fungi (AMF). At 55 dap, the soil chelate NTA was applied. The plants were subjected to four different experimental treatments: (1) 1 mM NTA; (2) 2 mM NTA; (3) 5 mM NTA; (4) 10 mM NTA. Plants were harvested 71 dap as soon as Pb toxic effect was observed. Plant foliage was dried in an oven at 65 °C for 48 h prior to acid digestion.

### 4.6. Effect of NTA and Combined Application of APG or Nano-Silica on Pb Phytoextraction

Switchgrass seeds were germinated (as described above) and transplanted into 300 mL grow-tubes containing the Pb-contaminated soil (see above). Plants were grown under controlled conditions (described above) in the Kennesaw State University Research Greenhouse. In this study, chemically enhanced phytoextraction using the chelator nitrilotriacetic acid (NTA) with and without nano-silica or the alkyl polyglucoside (APG) Triton X-100 (TX100) was tested [35,75]. The plants were subjected at 56 dap to four different experimental treatments: (1) Control; (2) NTA; (3) NTA + nano-silica; and (4) NTA + APG. The NTA (5 mmol kg^−1^ soil) solution was made by dissolving powdered NTA with 10 N NaOH solution. The Triton X-100 (2%) was added to the NTA + APG solution. A complete nutrient solution (described above) was supplied to all plants twice a week for 8 weeks. Soil-fungicide (propiconazole) was given to all plants for two weeks for suppression of symbiotic arbuscular mycorrhizal fungi. The nano-silica (500 mg kg^−1^ soil) was added to the NTA+nano-silica solution and was given to target plants twice a week for a one-month period. Plants were harvested at 72 dap. Plant foliage was dried in an oven at 65 °C for 48 h prior to acid digestion.

### 4.7. Acid Digestion and Chemical Analysis of Plant Samples

Dried plant material was digested using the HotBlock digestion system (Environmental Express^®^, Inc.) following a modified EPA method [76]. Dried plant tissues (500 mg) were digested in 38% HCl (10.0 mL) and 70% HNO_3_ (10.0 mL) in Environmental Express^®^ 100.0 mL plastic digestion tubes following a modified EPA Method 3050 B. The tubes were capped and rested at room temperature for 24 h, then refluxed at 95 °C in an Environmental Express^®^ HotBlock system for 55 min. Samples were capped and allowed to cool before their volume was brought to 100 mL using trace-metal grade DI H_2_O before being vacuum-filtered through 0.45 micron filterers and placed into 50 mL centrifuge tubes for storage in a refrigerator. Samples were analyzed using ICP-OES (Perkin Elmer Avio 200).

### 4.8. Statistical Analysis

Data were statistically analyzed using one-way analysis of variance (ANOVA) followed by post hoc Fisher’s Least Significant Difference (LSD) using IBM SPSS Statistics 27. Additionally, a relationship between categorical treatments and foliage Fe concentrations was calculated by regression analysis. Statistical significance was accepted at the level of *p* < 0.05.

## 5. Conclusions

Phytoextraction of Pb by switchgrass, enhanced with chelates, has many implications for both phytoextraction and the bioenergy industry. This study demonstrated that the phytoextraction of Pb by switchgrass was significantly enhanced using the combined chemical chelator NTA (5 mM) + APG (2%). Furthermore, this study demonstrated that the chelating agent NTA could achieve similar effects to EDTA and could therefore potentially replace EDTA. The use of EDTA in phytoextraction has met some concerns because of its long persistence time in soil and potential mobilization into groundwater. These results will be of benefit in the phytoextraction of Pb-contaminated soils and could potentially increase the commercial application of this technique. In addition, the merits of Pb phytoextraction extend beyond the removal of excessive heavy metals from soils. Currently, switchgrass biomass is harvested as a lingo-cellulosic biofuel feedstock for ethanol production and such cultivation could involve phytoextraction of contaminated lands.

## Figures and Tables

**Figure 1 plants-11-01012-f001:**
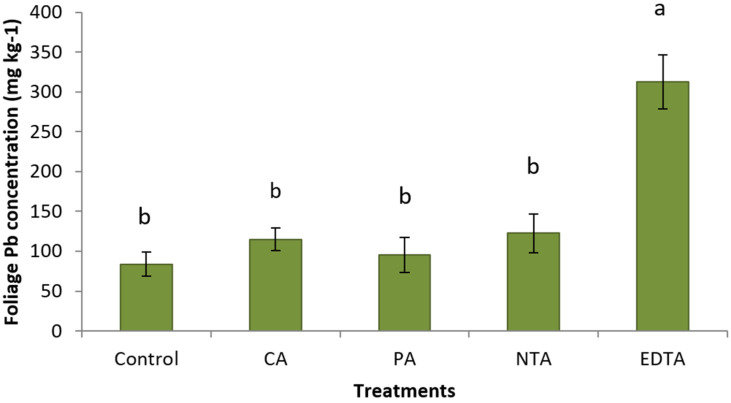
Average (±SD) foliage Pb concentration (mg kg^−1^) of *Panicum virgatum* at time of harvest. The means for columns designated with the same letter are not statistically significantly different (α = 0.05). Treatments labeled: CA = Citric Acid 10 mM; PA = Phytic Acid 10 mM; NTA 1 mM and EDTA 1 mM.

**Figure 2 plants-11-01012-f002:**
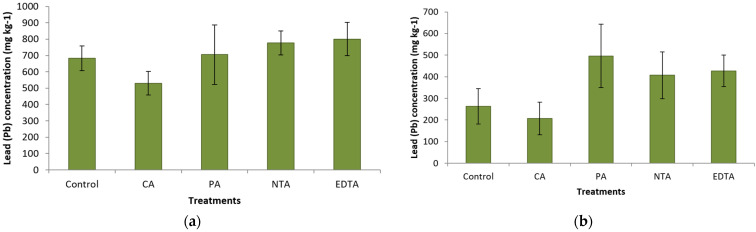
(**a**) Average (±SD) Pb concentration (mg kg^−1^) of soil wash with 0.05 M acidic acid glacial. (Treatments labeled: CA, Citric Acid; PA, Phytic Acid; NTA and EDTA). (**b**). Average (±SD) Pb concentration (mg kg^−1^) of soil wash with 0.01 M acidic acid glacial. (Treatments labeled: CA, Citric Acid; PA, Phytic Acid; NTA and EDTA).

**Figure 3 plants-11-01012-f003:**
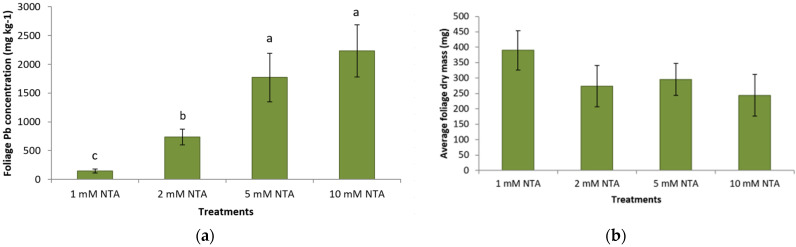
(**a**) Average (±SD) foliage Pb concentration (mg kg^−1^) of *Panicum virgatum* at time of harvest. The means for columns designated with the same letter are not statistically significantly different (α = 0.05). Treatments were different levels of NTA. (**b**) Average foliage DM (g) (±SD) of *Panicum virgatum* at time of harvest. The means for columns with same letter are not significantly different (α = 0.05). Treatments were different levels of NTA.

**Figure 4 plants-11-01012-f004:**
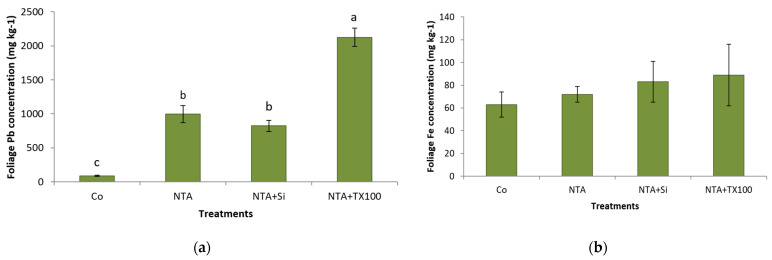
(**a**). Average (±SD) foliage Pb concentration (mg kg^−1^) of *Panicum virgatum* at time of harvest. The means for columns designated with the same letter are not statistically significantly different (α = 0.05). (Treatments labeled: Co, control; NTA, NTA 5 mM; NTA+Si, NTA 5 mM + nano-silica, NTA+TX100, NTA 5 mM + Triton X-100). (**b**) Average (±SD) foliage Fe concentration (mg kg^−1^) of *Panicum virgatum* at time of harvest. The means for columns designated with the same letter are not statistically significantly different (α = 0.05). (Treatments labeled: Co, control; NTA, NTA 5 mM; NTA+Si, NTA 5 mM + nano-silica, NTA+TX100, NTA 5 mM + Triton X-100). (**c**) Average (±SD) foliage Al concentration (mg kg^−1^) of *Panicum virgatum* at time of harvest. The means for columns designated with the same letter are not statistically significantly different (α = 0.05). (Treatments labeled: Co, control; NTA, NTA 5 mM; NTA+Si, NTA 5 mM + nano-silica, NTA+TX100, NTA 5 mM + Triton X-100).

## Data Availability

Not applicable.

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
