# Peer review of "Evaluation of Chelating Agents Used in Phytoextraction by Switchgrass of Lead Contaminated Soil"

_plants, 2022, doi:10.3390/plants11081012_

Round 1
Reviewer 1 Report
Materials and methods
L 232, I consider that there is too little bibliographic source considering that it is specified that the plant used has a high tolerance to lead.
The soil was polluted only with lead, iron, and al? Were there no analyzes of the concentration of other pollutants?
Why were soil samples not taken even at depths greater than 20 cm?
How deep is the plant root? Certain characteristics of the plant are not specified.
There is a lack of information between sampling the soil and filling the pots with soil. Can't explain what type of soil was added to the pot? How was the soil homogenized? Did all the pots have the same type of soil? After soil samples were taken, what happened to the soil? how the soil was prepared for experiments.
L 267: It is specified that the humidity has been kept constant, what is the percentage?
Why has the light period been supplemented? Does it influence plant growth? Will this be possible in the field/pilot ladder?
What are “a” and “b” in Figures 1, 3, 4? Figure 4 also shows the letter "c". Some letters do not specify these letters.
I recommend using the same ppm or mgkg-1 unit of measurement. In some figures, ppm appears, and in others mgkg-1
why weren't the same investigations done for all 3 metals with which the soil is polluted? If only PB appears in the title, why were Al and Fe also studied? Do these metals exceed the values ​​allowed by law?
Maybe it was good to divide the 3 investigated metals on how to extract them from the soil by using different chelating agents.
On the discussion side, the investigations carried out on Fe and Al are not mentioned at all.
References: Bibliographic sources are not inserted in the text in square brackets.
I recommend a few bibliographic sources to be introduced in the introductory part, where it is remembered that there are several extraction agents. 4 are specified in the paper, it must be specified that there is not only this.
- A. M. Chirila Babau, V. Micle, G. E. Damian, I. M. Sur. Preliminary investigations on the potential of Robinia pseudoacacia L. (Leguminosae) in phytoremediation of waste dumps. Journal of Environmental Protection and Ecology 21, No 1, 46-55 (2020), Soil Pollution
- GE GE Damian, V. Micle, IM Sur, Removal of Heavy Metals from Contaminated Soils Using Chitosan as a Washing Agent - A Preliminary Study, Journal of Environmental Protection and Ecology 21, No. 3, 823–829 (2020), Soil pollution
- Gianina Elena Damian, Valer Micle, Ioana Monica Sur, Mobilization of Cu and Pb from soils contaminated with several metals by dissolved humic substances extracted from leonardite and factors affecting the process, Journal of Soils and Sediments, vol 19 (7), p . 2869-2881, 2019, https://doi.org/10.1007/s11368-019-02291-w.,
Author Response
My comments are in the attached file

Reviewer 2 Report
Evaluation of Chelating Agents Used in Phytoextraction by Switchgrass
of Lead (Pb) Contaminated SoilBy Greipsson et al.
This study investigated the efficiency of chelants (e.g. EDTA NTA, citric acid, and phytic acid) on phytoextraction of soil from switchgrass. Contamination of soil by Pb has become a common environmental issue not only in lead mining areas but everywhere as Pb is an important component of automobile parts, batteries, etc. Therefore, remediation of soil polluted by Pb is a priority task in soil management approaches. Phytoextraction is one of the soil remediation techniques that is used to accumulate the trace elements in plant parts such as shoots, roots, foliage, etc. Some of the phytoextraction techniques use chelants to enhance the accumulation of trace metals in plant parts. The problem with these chelants is that they can increase the risk of metal leaching in soil due to increased metal mobility by them. Therefore, finding a suitable chelant is a challenge. This study is one such study that evaluated the efficiency of various chelants. However, similar studies on Pb phytoextraction using similar chelants can be seen in the literature. The manuscript has several flows including lack of flow and structure, unclear sentences, poor presentation of data. It is important to highlight at the beginning which Pb fraction in soil that plants would uptake i.e. bioavailable/exchangeable fraction. As Pb in the soil can exist in many forms some of which are not bioavailable in their current form but can become bioavailable in the future. Also, different parts in the plants accumulate a different amount of Pb, here, only the foliage of switchgrass was tested for Pb extraction by different chelants. Would have been good to check the Pb accumulation in roots and shoots. The scientific writing in the manuscript needs to be improved. Some comments are given below and in the annotated manuscript.
Introduction
Objective and Research Problem are not presented in this section and found in the first paragraph of the Discussion. Similarly, some sections that should be included in the Introduction are in the Discussion. Authors need to highlight that fast degradation of chelant is an important factor when using those in the phytoextraction of trace metals. Some information on switcthgrass should be in the Introduction in addition to Materials and Methods.
Materials and Methods
Written fairly well, however at some places lacks flow also, lacks important information on soil sampling procedure.
Also difficult to understand how many experiment sets were conducted and their procedure. In the Abstract it is clear that there were 3 sets of experiments. It is important to highlight which Pb fraction was extracted by Acetic Acid (exchangeable/soluble??)
Results
Overall Results/Data are poorly presented. For example, some figures have horizontal lines and some are not (Figure 3). Units are different in y-axes and Figure caption (Pb concentration in Figure 2 y-axis presented as “ppm” and in the caption, it is as “mg kg-1”). In Figure 3 axes are not shown. In some graphs, x-axes are not labeled. Some figures are having borders (Figure 2) and some are not
Results could have been split into three different sections explaining three different sets of experiments.
Discussion
As mentioned earlier first paragraph should be in the Introduction. The rest of the Discussion sometimes lacks a clear relation to the results of this study.
No Conclusion. Or it is not clear in the Discussion.

Author Response
My answers are in the attached file

Reviewer 3 Report
Although not innovative, since similar papers were already published, the work and the presented results are meaningful. Please see below some comments.
The lead symbol Pb can be removed from the title.
It is recommended that the definitions of EDTA and NTA should be added in line 13.
The abstract contains too many results turning the text confusing.
Please check/clarify the sentence in lines 21/22, and add the definition of APG.
I recommend avoiding the use of lumped references, such as (1-7), (8-13),… Please try to summarize the contribution of each reference. In addition, it is recommended the use of more recent works. Finally, some references can be eliminated.
Please move the definition of NTA from line 63 to line 57, where the acronym was first mentioned.
Please define PA (Phytic acid) in line 87 or avoid the acronym.
Please check carefully the proper location of all acronym definitions and avoid repetitions throughout the text.
Please check the statistical significance of label (b) in Fig. 3.a.
I recommend adding statistical comparison of results in Fig. 2.a, 2.b, 3.bx, 4.a, 4.b
Please elaborate on the purpose of the soil washing experiment, clarifying why it was done after the plant had grown.
Please justify the option for using different growth conditions in the experiments of section 4.3 (5L pots), in comparison with the experiments described in sections 4.5 and 4.6 (300 mL tubes).
Please clarify how the soil samples were managed. Were the several samples mixed or used unmixed for the experiments?
I recommend that authors try to improve the description of methods to improve the clarity of the document. In addition, all analytical methods must be referred to.
Author Response
My answers are in the attached file

Round 2
Reviewer 2 Report
please see my comments in the annotated manuscript.
Conclusion should be restructured highlighting key results.

Author Response
Please see my corrections in the attached file.

Reviewer 3 Report
I acknowledge the authors efforts to made the proposed changes. I have no further comments.
Author Response
We would like to thank reviewer 3 for useful comments on our manuscript.